# Built environment as a risk factor for adult overweight and obesity: Evidence from a longitudinal geospatial analysis in Indonesia

Alka Dev[1]*, Jennifer Brite[2], Frank W. Heiland[3], Deborah Balk[4]

1 The Dartmouth Institute for Health Policy and Clinical Practice, Geisel School of Medicine at Dartmouth College, Lebanon, New Hampshire, United States of America, 2 York College (Public Health), CUNY Institute for Demographic Research, City University of New York, New York, New York, United States of America, 3 Marxe School of Public and International Affairs, The Graduate Center of CUNY (Economics), Associate Director, CUNY Institute for Demographic Research, City University of New York, New York, United States of America, 4 Marxe School of Public and International Affairs, The Graduate Center of CUNY (Economics, Sociology), CUNY Institute for Demographic Research, City University of New York, New York, United States of America

* alka.dev@dartmouth.edu

**Data Availability Statement:** The dataset supporting the conclusions of this article are available in the Open Science Framework repository: https://osf.io/23adg/.

## Abstract

Indonesia has nearly doubled its urban population in the past three decades. In this period, the prevalence of overweight and obesity in Indonesia has also nearly doubled. We examined 1993–2014 panel data from the Indonesian Family Life Survey (IFLS) to determine the extent to which the increase in one's built environment contributed to a corresponding increase in adult overweight and obesity during this period. We estimated longitudinal regression models for body mass index (BMI) and being overweight or obese using novel matched geospatial measures of built-up land area. Living in a more built-up area was associated with greater BMI and risk of being overweight or obese. The contribution of the built environment was estimated to be small but statistically significant even after accounting for individuals' initial BMI. We discuss the findings considering the evidence on nutritional and technological transitions affecting food consumption patterns and physical activity levels in urban and rural areas.

## Introduction

Indonesia has seen substantial improvements in development and health indicators in the past three decades. Since 1993, the standard of living, measured as per capita real gross domestic product (GDP), has doubled from USD $4,800 to USD $11,100 [1]. The infant mortality rate has fallen from 62 deaths per 1000 live births in 1990 to 20 in 2019, while maternal mortality has fallen from 272 deaths per 100,000 live births in 2000 to 177 in 2017 [2, 3]. Life expectancy has risen from 62 years in 1990 to 72 years in 2019 and the share of the urban population has grown from 30.5% to 56.0% [4, 5]. Survey data show that the proportions of overweight and obese individuals have also doubled since the mid-1990s, among both men and women [6]. Rising incomes and technological advances are contributing to this development. Studies using cross-sectional Demographic Health Survey (DHS) data in low and middle-income

**Funding:** The authors received no specific funding for this work.

**Competing interests:** The authors have declared that no competing interests exist.

countries (LMICs) found a positive association between BMI and per capita GDP with the middle classes in the richer countries having the highest odds of being overweight or obese [7, 8]. Rates of overweight and obesity were also higher among urban women compared to rural women and among those who were wealthier [9]. However, given the reliance on cross-sectional data in LMICs, it is not always feasible to examine the relationship between urbanization and overweight or obesity outcomes over time in large rapidly developing countries such as Indonesia.

The main feature of development in Indonesia during the past three decades has been rapid urbanization. Jakarta has become one of the world's largest cities and while it operates as the core urban center of the country, it is only one of several large cities and towns in Indonesia with populations over one million people [10]. With urbanization playing a key role in Indonesian life, it is important to find more sophisticated methods to measure urban conditions than the typical urban/rural dichotomy often found in surveys. For example, our study in India found a strong, positive city-size gradient with respect to proportions overweight [9]. One particular data product, the Global Human Settlement Layer (GHSL) described in detail below, is available as a time series covering the period 1975 to 2014, making it ideal to consistently describe the degree of built-up density as a proxy for urbanization [11]. Since most surveys capture urbanization as a single stratum only—urban vs. rural—new datasets that allow us to determine a fuller urban continuum are important for capturing recent changes and understanding the urban demographic future, which is of great interest to both researchers and policymakers in global health and development [12].

Using longitudinal data from the Indonesian Family Life Survey (IFLS), which spans the period 1993 to 2014, we investigated whether urbanization, as measured by increases in the built environment, could explain the rise in overweight and obesity in Indonesia during the past three decades. The IFLS data provide a unique window into this key period of modernization in Indonesia. High-quality panel data like the IFLS are rare, especially in the developing world. We used four waves of the IFLS, allowing us to follow the weight trajectories of individuals for 21 years in large representative samples. The longitudinal nature of these data permits statistical inferences that are more robust to individual heterogeneity than evidence from (repeated) cross-sections such as the DHS or from treating IFLS panels as cross-sections.

To advance the analysis of the role of urbanization, we supplemented the IFLS urban-rural designations with satellite-based measures of the proportion of land area that is built up over the period of the survey, which we spatially matched to the respondent's survey cluster location. The Global Human Settlement Layer database defines built-up as manmade objects including buildings, associated structures, and civil works [13]. We refer to these alternative measures as indicators of urbanization, which describes conditions that are particular to an urban setting or that are found to a much greater extent in urban areas [14]. These longitudinal data allow us to examine the determinants of obesity in the context of a developing country undergoing rapid development. This is of scientific interest because relatively little is known about the relationship between changes in the built environment and individual BMI outcomes during times of economic and urban transition, accounting for lagged exposures. We also analyzed how our measure of land use compares, in terms of its explanatory contribution, to using a binary urban-rural measure.

The remainder of the paper is organized as follows: In Section II, we briefly discuss the related theoretical and empirical literature. Section III describes the IFLS data, the choice of sample, and the measures used in the analysis. Section IV describes the analytical approach and presents evidence from basic descriptive and multivariate analyses. In section V, we discuss the findings and conclusions.

## Materials and methods

### Data and sample selection

**Indonesia Family Life Survey (IFLS)—panels 1–5.** The IFLS spans 21 years of Indonesian economic development between 1993 and 2014. It is a national longitudinal survey in Indonesia, consisting of five panels: 1993, 1997, 2000, 2007–08, and 2014. The first panel was representative of approximately 83% of the Indonesian population, covering 13 of 27 provinces. The initial sampling frame was stratified on provinces and enumerated based on a prior national survey and census of which 321 enumeration areas (EAs) or clusters were randomly selected. The final sample included 7,224 partially or fully completed households; 48% of which were urban. Re-contact rates across panels were greater than 90%. To maintain an equal duration of seven years between panels, we utilized data from the 1993, 2000, 2007, and 2014 panels.

We included all respondents ages 18 or over who were assessed for health measurements in all four waves and who were never underweight, resulting in a sample size of 3,770 men and women. We excluded underweights as weight gain in this subgroup would largely confer a health benefit (n = 1,345). The restriction of complete panel data across the 21 years (4 waves) resulted in disproportionally dropping older individuals and men (as of 1993). Further analysis showed that those dropped tended to be from more urban and built-up areas but had comparable BMI values (by sex) to those who were included in all four panels.

Two trained nurses assessed all individuals for health measurements during the survey, unless participants were too ill or pregnant, as determined by them at the time of the interview. Height was measured in centimeters and weight was measured in kilograms. Biologically implausible (<100 or >198.8) or missing heights were replaced with heights from other waves where possible while people with biologically implausible weights (<20 or >167.6) were removed from the sample altogether. BMI was calculated as weight in kilograms divided by height in meters squared ($kg/m^2$). Pregnant women were excluded from the sample. Respondents were also excluded from the analysis if they were missing height or weight data or had biologically implausible BMIs (<10 or >50).

The geographic location of each survey cluster (i.e. a collection of households in the sampling frame) was made available for restricted use.

**Global Human Settlement Layer (GHSL) dataset.** We used data from the Joint Research Council (JRC) of the European Commission's GHSL Built-Up Grid Project which integrates existing information on global human settlement with new information extracted from available remote sensed (RS) imagery, largely Landsat, on above-ground buildings [15]. A newly released data product, built-up area is defined as any given area (at 38-meter resolution) where more than 50% of the area contains above-ground buildings. The GHSL definition of buildings includes both permanent and temporary structures. An aggregated data product, at roughly 300 meters resolution, sums the dichotomous 38-meter pixels to generate a measure of the percentage of land area that is built up. Higher values of built-up area serve as a measure of higher levels of urbanization. These data are increasingly being used as proxies for urbanization [16, 17]. The GHSL classification schema does not assume any embedded urban/rural dichotomy.

The dataset supporting the conclusions of this article is available in the Open Science Framework repository [18].

### Analysis

Our analytical strategy (described in greater detail in Section IV below) consisted of a series of regressions predicting an individual's BMI and risk of being overweight or obese based on

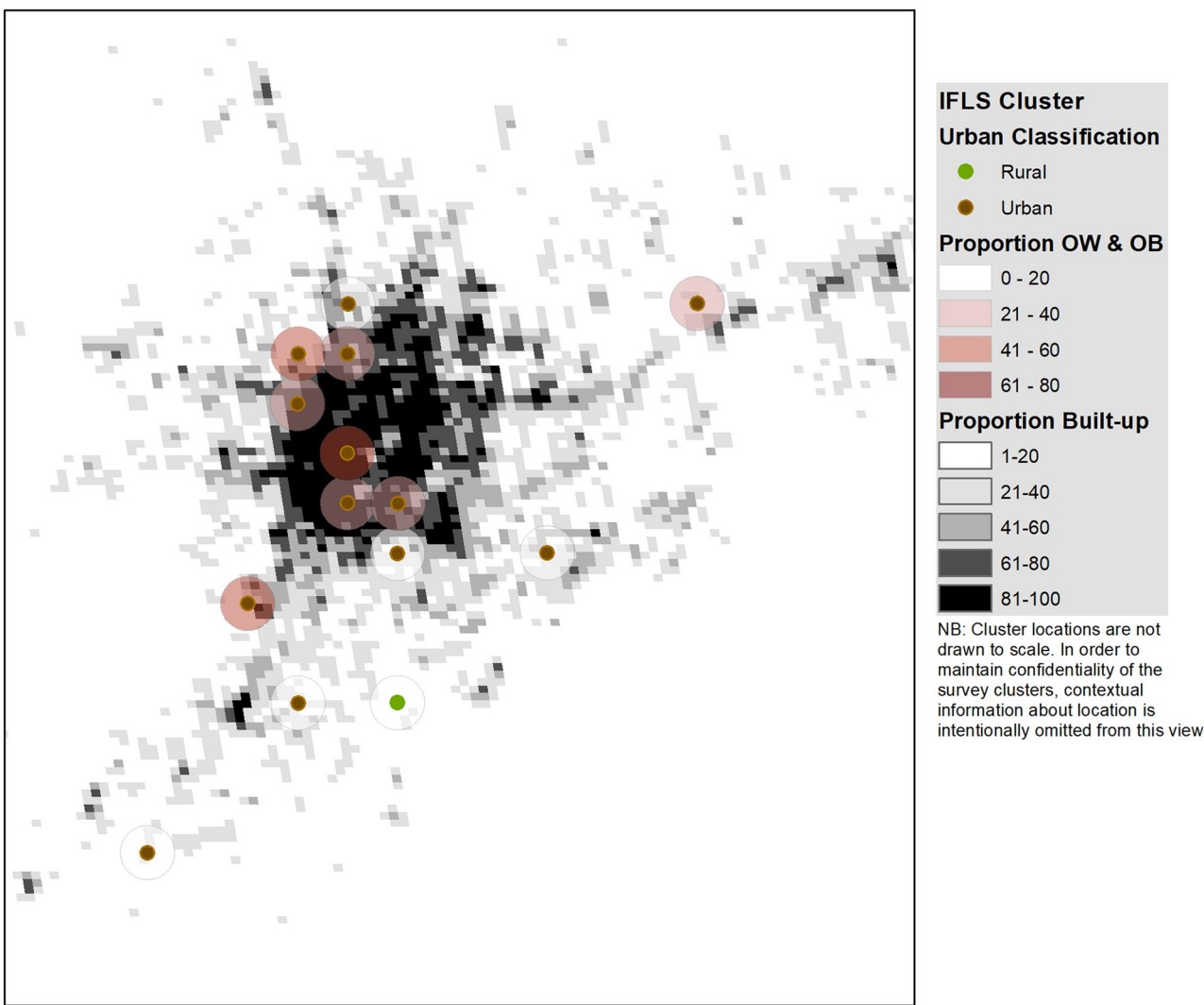

**Fig 1. Illustration of data integration: Survey clusters and built-up area (stylized view).**

individual/household-level and geographic determinants of weight. We exploited the variation in weight outcomes and weight determinants that exists between individuals as well as within individuals (over time) in our four-panel dataset. Fig 1 illustrates the data layers of interest, described below: urban-rural strata as determined by the survey team, built-up area from GHSL, and the cluster-level proportion of overweight or obese (because individual-level BMI cannot be rendered on the map) with a 2km buffer around each cluster.

**Dependent variables.** Our main outcomes of interest were individual BMI and the risk of being overweight or obese. We used continuous BMI and also constructed a binary variable of whether a person was overweight or obese versus normal weight for each panel. The following cut-offs were used for analysis with BMI as a categorical outcome: 18.5–22.9 (normal), 23–29.9 (overweight), and > 30 (obese). Overweight is generally defined as a BMI of 25kg/m$^2$ or higher while obesity is defined as having a BMI of 30kg/m$^2$ or higher [19]. However, a BMI cutoff of 23 kg/m$^2$ has been recommended for Asian populations who might be at higher risk of type 2 diabetes and cardiovascular disease at lower BMIs than the existing WHO cut-off point of

$25kg/m^2$ for overweight [20]. To accommodate for BMI threshold differences among Asians, we used the overweight cut-off recommended for Asian populations in our analysis [21].

**Predictor variables.** *Dichotomous urban-rural*. IFLS included a binary measure based on the National Socioeconomic Survey (SUSENAS), a nationally representative survey that distinguished between urban and rural areas based on five criteria: 1) Population density, 2) Proportion of agriculture households, 3) Access to urban facilities (schools, market/shops, hospital, cinema, hotels/motels, % of household using telephone, % of household using electricity), 4) Availability of public supporting facilities (main street lighting, commercial banks, public phone), and 5) Proportion of land used for other than housing [22]. The sampling frame used in the 1993 IFLS utilized a scoring system based on the first three items in the criteria listed above while the last two items were introduced in the 2000 SUSENAS. The IFLS redefined urban and rural with each panel.

*Built-up area*. GHSL data are available for 4 periods centering around our target years: 1975, 1990, 2000, and 2014, making it the first-ever spatial layer indicating change over time in built-up areas. We examined several different radii when conducting data analysis (2, 5, and 10km) and found the results to be not qualitatively different. We chose 2 km because we felt conditions most proximal to a respondents' home would have the greatest impact on obesity risk. Buffers were used to measure the built-up character of neighborhoods (proxying for urban) rather than simply the level of built-up at the cluster location; further, they were not intended to measure walkability, which would depend on many factors including the presence of dedicated pedestrian networks such as sidewalks or walking paths, which cannot be specified in these data [23]. Mean built-up raster values for the 2km IFLS clusters were calculated for 1990, 2000, and 2014 in ArcGIS 10.8 and then converted to percentage built-up [24]. Annual percent growth rates between 1990, 2000, and 2014 were used to impute built-up percentages in 1993 and 2007. Change in the percent built-up was also calculated for the periods 1993–2000, 2000–2007, and 2007–2014 for each cluster. Built-up percentage values were matched to individuals based on their IFLS cluster-ID. To get a sense of these inputs, Fig 2 shows the changes in built-up area in Indonesia, with provincial boundaries, between 1990 and 2014. Fig 3 zooms in to show two anonymized locations–one of a small urban area or town that we call less built-up and another relatively more built-up area, at the same two points in time. Overlaid on this map are the IFLS cluster designations for urban-rural strata to show how dichotomous urban-rural designations may mask underlying urban characteristics–that is, a rural area that is proximate to highly built-up area (left-side panels) as well as an urban cluster in areas with little to no built-up land (right-side panels).

Because we made use of novel data typically not used in health and behavioral studies, we pause to describe the built-up measure we adopted. As shown in Figs 2 and 3, even small cities achieve high levels (80% or higher) of built-up area at their cores. Our analysis does not consider other spatial characteristics such as total built-up area in the vicinity of the indexed cluster, which could approximate city-size, or connectivity of the indexed location to nearby ones: that is, we do not distinguish between a 40% built-up area cluster that is on the outskirts of Jakarta from 40% built-up area cluster on the periphery of a much smaller city or town [25].

Fig 4 boxplots further depict the heterogeneity of urban clusters by panel-specific classification. For each panel, the distribution of percentage built-up (y-axis) is shown by urban-rural strata of the cluster (x-axis) with the interquartile range (25–75%) shown in the box, and the median value shown by the line within the box. Rural locations have a concentrated and low-level built-up distribution. The median built-up percentage is less than 5% in all panels, though the interquartile increases from about 10 to 15 percentage points from 1997 to 2014. Urban locations, in contrast, have a much greater median percent built-up, approximately 40% in all panels, and much greater dispersion or heterogeneity in the degree built-up of places classified

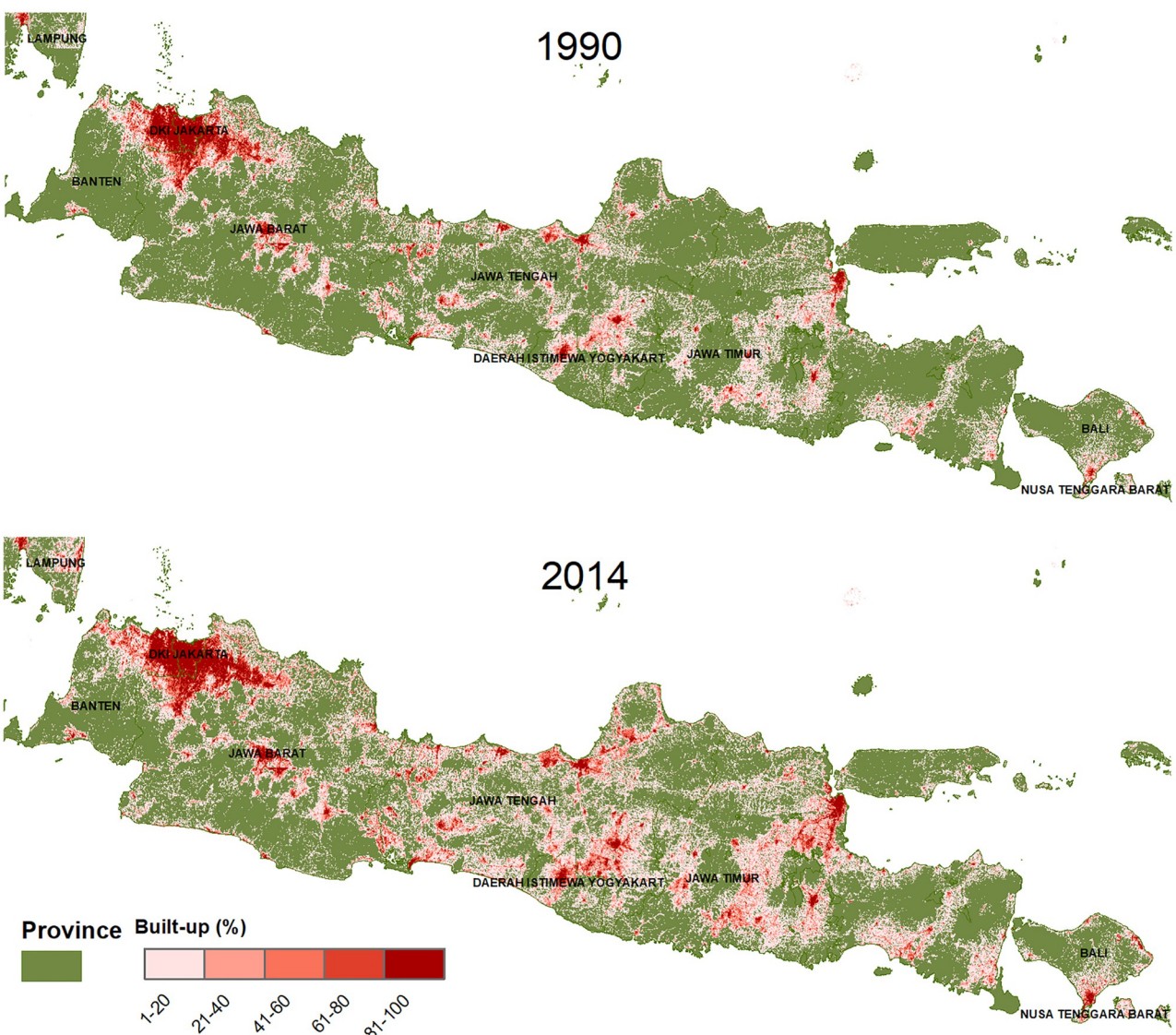

Source data: Pesaresi et al., 2016 available from https://ghsl.jrc.ec.europa.eu/ghs_bu2019.php

**Fig 2. Indonesia provinces and Global Human Settlement Layer (GHS) data, Built-up Land Area (%), Java close-up, 1990–201.** Source data: Pesaresi et al., 2016 downloadable from: https://ghsl.jrc.ec.europa.eu/ghs_bu2019.php.

as urban. Neither the median nor the range change much over the 21-year observation period although there is a small increase in median urban built up and a widening of box for rural clusters suggesting, as expected, that all areas are getting more built-up over time.

*Other location-based measures.* We also constructed binary indicators for residence on the island of Java, Sumatra, and "Other" to account for the potential confounding from Java–the largest, most urban, and most densely populated island.

**Covariates.** We included commonly-used socio-economic, cultural, and health behavior variables as covariates [26–28]. Age and age-squared at the time of each survey were introduced as continuous variables in every model. Age was self-reported, and we restricted our sample to 18 years and older at time of first panel. Education level at the time of the survey was introduced as a categorical variable and included the following six groups: none, elementary,

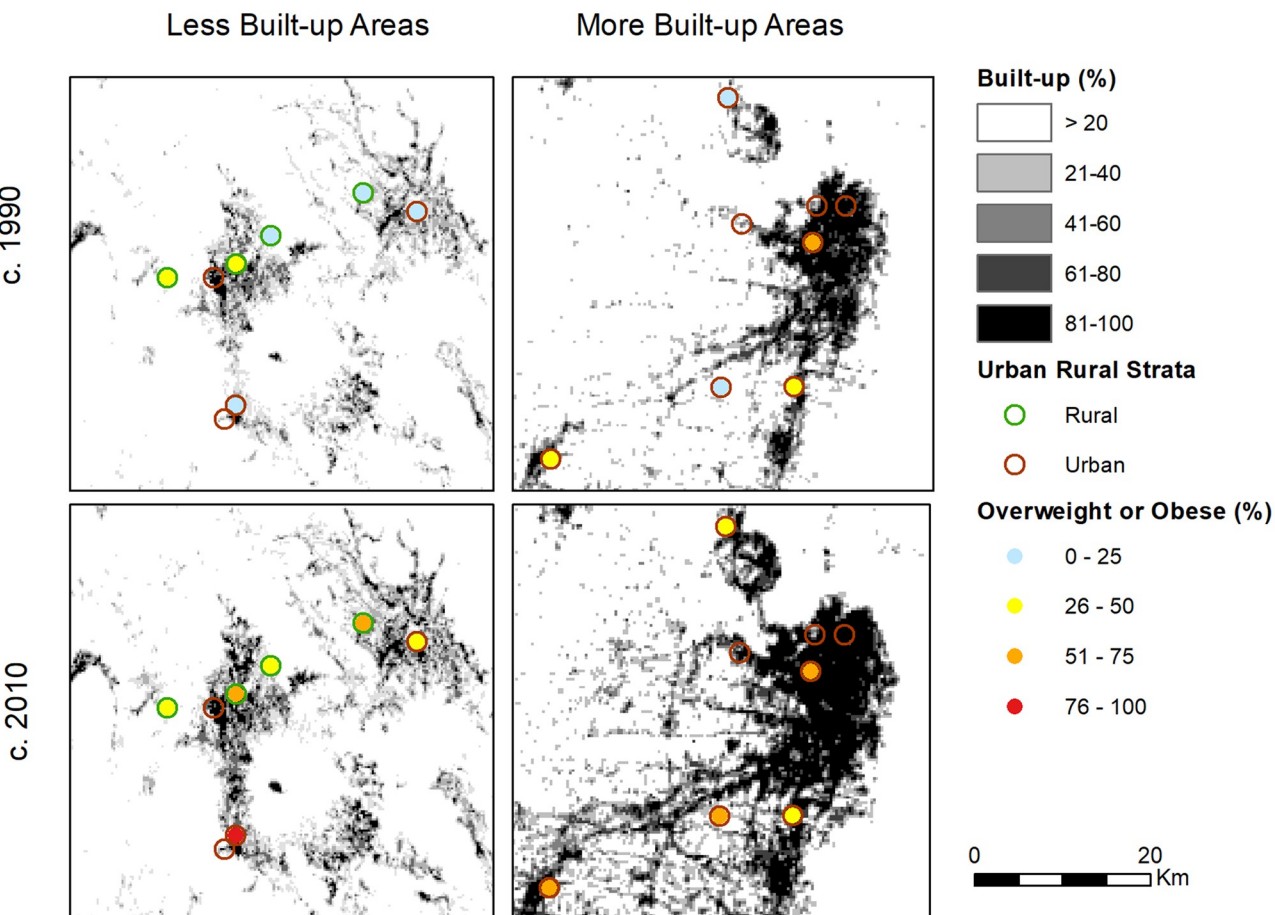

**Fig 3. Built-up area close-ups with example, anonymized cluster locations, IFLS.** NB: Maps are shown without identifying information to protect the confidentiality of survey respondents. Percent of cluster population that is overweight or obese is shown as this is the dependent variable, though the unit of analysis is individual not cluster as shown. Percentages are not given for any cluster with fewer than 10 respondents in any of the four panels.

junior high, senior high, college or higher, and other. The 'other' category generally included vocational and religious schools. Across waves, the educational questions became more detailed regarding schools outside the standard educational system (such as Muslim schools and schools for the disabled), resulting in an increase in the other category, particularly in the most recent survey.

Marital status at the time of the survey was introduced as a categorical variable and included the following three groups: never married, married, and widowed or other. Religion at the time of the survey was introduced as a categorical variable and included the following three groups: Muslim, Christian, and Hindu/Buddhist/other. Smoking was recorded as whether a person was a smoker at the time of the survey or not. Dummy variables were created for all missing values for any covariate, and missing data were re-coded to zero. Additional covariates of interest, such as income, occupation, and food expenditures were not consistently available or viable to use for all four panels, and regrettably, we cannot account for these possible mechanisms. We did not consider migration for work to be a significant issue in the sample as the cohort was interviewed and BMI measured over several panels at the same location.

**Modeling approach.** Utilizing the longitudinal nature of the IFLS data, we estimated a series of multivariate regressions using two types of model specifications: "total input" models

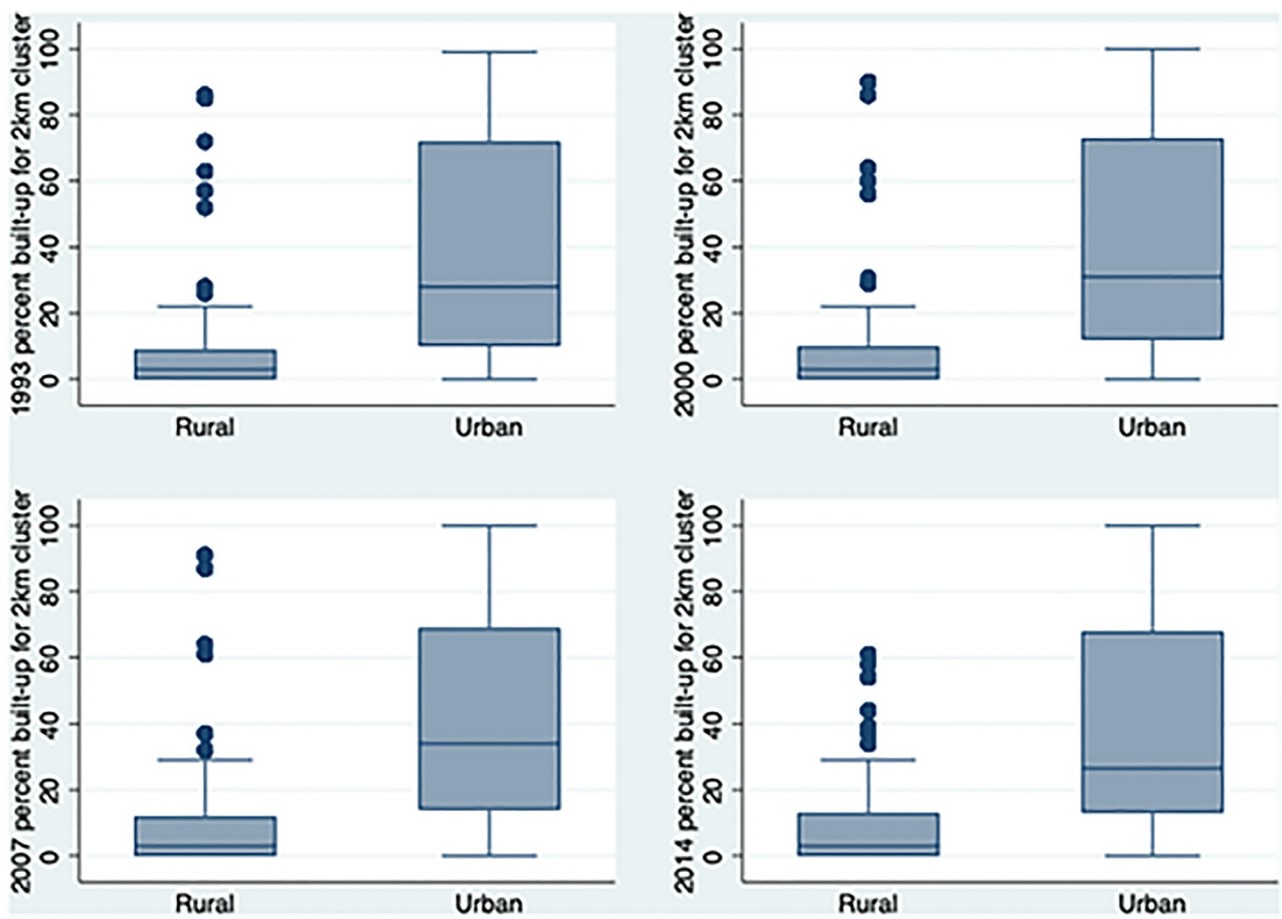

**Fig 4. Boxplots of percent built-up by urban-rural clusters, 1993–2014 (IFLS).**

and "value-added" models. The idea behind the total input approach is to account for all determinants of weight including any relevant current and past exposures, as well as weight-related preferences and endowments. In our total input models, we predicted individuals' weight outcomes in a given survey year based on individual-level, household-level, and geographic variables. Since the IFLS data—like most data sets—are limited in terms of input variables, especially past exposures, the total input models faced the risk of being mis-specified due to omitted factors. Therefore, in the value-added models, we specified current BMI status as a function of BMI in the previous wave (7 years earlier), along with the same set of individual/household-level and geographic factors. The idea was that lagged BMI would account for any unobserved (earlier) determinants of weight, resulting in more conservative estimates of the impact of built-up on BMI and overweight/obesity [29].

Specifically, we pooled the panel data from four IFLS waves to obtain three 7-year periods for each respondent: 1993–2000, 2000–2007, and 2007–2014. In each period, individuals' BMI at the end of the period was modeled with contemporaneous measures of age, age squared (divided by 100), education, religion, marital status, and smoking. We examined the role of physical built-up with two strategies: In one set of specifications, we used contemporaneous built-up ("% built-up area of current residence"). In another, we used 7-year lagged built-up ("% built-up area of residence in previous panel") and change in built-up between waves ("change in % built-up area since previous panel"). The latter specification is a generalization

of the former since it is mathematically equivalent to a specification with current *and* lagged built-up.

The same model specifications were then estimated using a binary overweight/obese variable as the outcome to estimate the (linear) probability of being overweight/obese at the end of each period. The value-added models included individuals' (7-year) lagged BMI values. In addition to models combining men and women, we also estimated models stratified by sex (shown in S1–S8 Tables). Each model was evaluated for best fit using the overall $R^2$ statistic.

### Ethics statement

This study was reviewed and exempted by the Institutional Review Board at Baruch College of the City University of New York (IRB File #2015–0426).

## Results

### Sample characteristics

**Sample means and proportions.** Tables 1 and 2 provide means and proportions of key variables in our longitudinal analytic sample by survey panel, sex, and rural-urban residence. Overall, there were significantly more women (n = 6,918 or 61.2%) than men (n = 4,392 or 38.8%) in our panel. As shown in Table 1, on average, individuals in our panel were 36 years old in the first wave (1993). Individuals were of age 18 and above at baseline, with the vast majority being concentrated between 18 and 59 years, as shown in Table 2. This reflects our age restriction as well as the fact that the survey sought to sample one couple over 50 years in each household, if possible. Those residing in areas classified as urban in the IFLS tended to be slightly older compared to those in rural areas. Men were about 2 years older than women in both rural and urban areas across all waves.

As shown in Table 1, mean BMI values rose steadily over the 21-year observation period from 22.4 in 1993 (IFLS1) to 24.8 in 2014 (IFLS5). Women in every panel had mean BMIs that exceeded those of men, regardless of urban or rural residence; this sex-differential became more pronounced after the first wave. Consistent with the trends in BMI, the proportion of panel members classified as (Asian) overweight or obese rose dramatically from 33.5% in 1993 to 62.8% in 2014 (see Table 2). The biggest jump in the proportion overweight or obese occurred between 1993 and 2000. More urban and rural women were classified as overweight or obese than men; up to nearly 1.5 times more in urban areas and up to twice as many in rural areas.

As Indonesians aged over the observed life span, they were also increasingly likely to live in more urban environments. From the sample sizes in Table 1, we can infer that the percentage of individuals residing in areas classified by IFLS as urban rose from 38.2% in 1993 to 53.3% in 2014. Consistent with rapid urbanization, average built-up percent in all locations increased steadily from 19.8% in the 1993 panel to 25.8% in the 2014 panel. As expected, the mean built-

**Table 1. Sample means by panel year, residence, and sex.**

| | 1993 | | | | | 2000 | | | | | 2007 | | | | | 2014 | | | | |
|---|---|---|---|---|---|---|---|---|---|---|---|---|---|---|---|---|---|---|---|---|
| | Urban | | Rural | | Total | Urban | | Rural | | Total | Urban | | Rural | | Total | Urban | | Rural | | Total |
| | F | M | F | M | | F | M | F | M | | F | M | F | M | | F | M | F | M | |
| **Mean BMI** | 23.5 | 22.4 | 22.3 | 21.5 | **22.4** | 24.9 | 23.1 | 23.5 | 21.9 | **23.1** | 25.8 | 23.8 | 24.5 | 22.6 | **24.3** | 26.2 | 24 | 25.2 | 22.7 | **24.8** |
| **Mean Age** | 36.4 | 38.5 | 36.2 | 38 | **37** | 43.6 | 45.6 | 43.2 | 45.1 | **44.1** | 50.7 | 52.8 | 50 | 51.8 | **51.1** | 57.4 | 59.5 | 57 | 58.8 | **58** |
| **Mean Built-up (%)** | 40.1 | 38.6 | 7.6 | 7.5 | **19.8** | 41.3 | 40.2 | 8.4 | 8.5 | **21.3** | 42.2 | 40 | 9.8 | 10.3 | **23.9** | 41.1 | 39.4 | 9.1 | 9 | **25.8** |
| **Observations (Persons)** | 911 | 529 | 1,395 | 935 | **3,770** | 953 | 544 | 1,353 | 920 | **3,770** | 1,048 | 620 | 1,258 | 844 | **3,770** | 1,255 | 755 | 1,051 | 709 | **3,770** |

**Table 2. Sample proportions by panel year, residence, and sex.**

| | 1993 | | | | 2000 | | | | 2007 | | | | 2014 | | | |
|---|---|---|---|---|---|---|---|---|---|---|---|---|---|---|---|---|
| | Urban | | Rural | | Total | Urban | | Rural | | Total | Urban | | Rural | | Total | Urban | | Rural | | Total |
| | F | M | F | M | | F | M | F | M | | F | M | F | M | | F | M | F | M | |
| BMI Group | | | | | | | | | | | | | | | | | | | | |
| **Normal** | 50 | 66.2 | 66.4 | 82.7 | **66.5** | 33.1 | 55.7 | 51.7 | 75.8 | **53.4** | 27.1 | 46.6 | 40.1 | 64 | **43** | 25.1 | 44.8 | 33.9 | 60.6 | **38.2** |
| **Overweight/Obese** | 50 | 33.8 | 33.5 | 17.3 | **33.5** | 66.9 | 44.3 | 48.3 | 24.3 | **46.6** | 72.9 | 53.4 | 59.9 | 36 | **57.1** | 74.9 | 55.2 | 66.1 | 39.4 | **61.8** |
| Age Group (years) | | | | | | | | | | | | | | | | | | | | |
| **18–24** | 8.3 | 3.2 | 11.4 | 4.8 | **7.9** | 0.1 | 0 | 0.1 | 0 | **0.1** | 0 | 0 | 0 | 0 | **0** | 0 | 0 | 0 | 0 | **0** |
| **25–29** | 16.6 | 1.2 | 18.5 | 14.1 | **16.1** | 4.3 | 1.3 | 6.5 | 2.4 | **4.2** | 0 | 0 | 0 | 0 | **0** | 0 | 0 | 0 | 0 | **0** |
| **30–34** | 21.4 | 21.4 | 19.1 | 22.1 | **20.7** | 11.8 | 6.3 | 16.2 | 10.2 | **12.2** | 1.8 | 1.5 | 2.8 | 0.8 | **1.9** | 0 | 0 | 0 | 0 | **0** |
| **35–39** | 20.9 | 22.7 | 15.4 | 18.4 | **18.5** | 21.5 | 17.5 | 20 | 20.1 | **20** | 9.1 | 3.7 | 13.5 | 6.4 | **9.1** | 0.6 | 0.4 | 0.3 | 0.4 | **0.4** |
| **40–44** | 12.8 | 17.4 | 14.4 | 16 | **14.8** | 22.3 | 26.1 | 16.8 | 21.5 | **20.7** | 19.2 | 14.2 | 20.5 | 16.6 | **18.5** | 6.3 | 2.8 | 9.5 | 2.7 | **5.8** |
| **45–49** | 7.5 | 10.5 | 6.2 | 10.1 | **7.9** | 14.9 | 18.8 | 15.5 | 15.8 | **15.9** | 20.4 | 23.2 | 18.9 | 22.2 | **20.8** | 15.7 | 10.6 | 17.3 | 13.3 | **14.7** |
| **50–59** | 11.5 | 10.1 | 12.2 | 10.1 | **11.5** | 18.4 | 22.4 | 16.2 | 20.1 | **18.6** | 30.7 | 35 | 25.8 | 33.2 | **30.3** | 40.8 | 41.1 | 35 | 41.9 | **39.4** |
| **60–69** | 1.7 | 3.9 | 2.4 | 4 | **2.3** | 6.1 | 7.4 | 7.2 | 7.9 | **7.1** | 15.5 | 17.4 | 14.3 | 13.7 | **15.0** | 22.1 | 28.2 | 22.4 | 28.5 | **24.6** |
| **70+** | 0.3 | 0.2 | 0.4 | 0.1 | **0.3** | 0.7 | 0.4 | 1.6 | 2.0 | **1.3** | 3.3 | 5.0 | 4.1 | 5.7 | **4.4** | 14.6 | 17 | 15.5 | 13.3 | **15.1** |
| Education level | | | | | | | | | | | | | | | | | | | | |
| **None** | 11.4 | 3.6 | 26.7 | 14.2 | **16.7** | 9.8 | 3.3 | 24.3 | 11.5 | **14.5** | 12.5 | 5.6 | 23.4 | 11.7 | **14.8** | 13.5 | 4.6 | 25.4 | 13.0 | **15.0** |
| **Elementary** | 55.8 | 50.7 | 58.4 | 62.6 | **57.7** | 56 | 49.4 | 60.8 | 64.5 | **58.9** | 53.6 | 48.5 | 60.8 | 63.7 | **57.5** | 52.3 | 50.1 | 55.8 | 60.9 | **54.4** |
| **Jr. High** | 15.3 | 16.6 | 8.3 | 8.4 | **11.2** | 15.1 | 17.1 | 8.1 | 9.1 | **11.4** | 15.6 | 16 | 8.7 | 8.3 | **11.7** | 13.7 | 13.1 | 6.1 | 7.6 | **10.3** |
| **Sr. High** | 15.1 | 20.4 | 5.5 | 12.6 | **11.7** | 15.6 | 21.5 | 4.4 | 10.7 | **11.2** | 13.2 | 20.6 | 3.8 | 10.3 | **10.6** | 9.9 | 18.8 | 3.3 | 6.8 | **9.3** |
| **College +** | 2.1 | 8.7 | 1 | 2.1 | **2.6** | 3.1 | 8.6 | 2.1 | 3.8 | **3.7** | 4.9 | 9 | 2.4 | 5 | **4.7** | 4.7 | 8.5 | 2.9 | 5.6 | **5.1** |
| **Other** | 0.3 | 0.0 | 0.1 | 0.0 | **0.1** | 0.3 | 0.0 | 0.3 | 0.3 | **0.3** | 0.2 | 0.2 | 1.0 | 0.9 | **0.6** | 5.9 | 4.9 | 6.5 | 6.1 | **5.9** |
| Marital status | | | | | | | | | | | | | | | | | | | | |
| **Never married** | 1.6 | 3.2 | 1.2 | 2.7 | **2** | 0.9 | 1.3 | 0.4 | 0.8 | **0.8** | 0.4 | 1.5 | 0.1 | 0.2 | **0.5** | 0.4 | 0.8 | 0.2 | 0.1 | **0.4** |
| **Married** | 91.5 | 96 | 91 | 96.3 | **93.1** | 86.4 | 96 | 87.4 | 97.6 | **90.9** | 78.1 | 94.4 | 79.5 | 96.6 | **85.4** | 66.7 | 90.2 | 71 | 93.5 | **77.6** |
| **Widowed and other** | 6.8 | 0.7 | 7.8 | 1.1 | **4.9** | 12.6 | 2.8 | 12.1 | 1.5 | **8.3** | 21.4 | 4.2 | 20.3 | 3.2 | **14.1** | 32.9 | 9 | 28.8 | 6.3 | **22** |
| Religion | | | | | | | | | | | | | | | | | | | | |
| **Muslim** | 91.1 | 90.2 | 86.7 | 84.4 | **87.7** | 90.6 | 89.7 | 88 | 84.9 | **88.1** | 90.7 | 87.4 | 87.1 | 86.1 | **88** | 91.2 | 88.3 | 86.3 | 84.8 | **88.1** |
| **Christian** | 5.5 | 5.1 | 5.2 | 5.3 | **5.3** | 5.8 | 5 | 5.2 | 5.3 | **5.3** | 5 | 4.8 | 6.1 | 5.8 | **5.5** | 4.3 | 3.8 | 6.6 | 6.6 | **5.3** |
| **Hindu, Buddhist, Other** | 3.4 | 4.7 | 8.1 | 10.3 | **7** | 3.7 | 5.3 | 6.9 | 9.8 | **6.6** | 4.3 | 7.7 | 6.8 | 8.1 | **6.5** | 4.4 | 7.8 | 7.1 | 8.6 | **6.6** |

up percentage was much higher in urban than rural areas, by a factor of four or more in each panel. Table 1 suggests that physical environment becomes more built up (at least until 2007) even in locations that remained classified as rural in the IFLS.

Table 2 also reports on three demographic characteristics: education, marital status, and religion. As expected, several respondents attained more education during the observation period. The majority of men and women had an elementary school education with junior or senior high as the next largest category. A greater proportion of men in urban areas had a senior high or college education. Most men and women were also married in both urban and rural areas but the proportions who were widowed or divorced rose throughout the observation period. As expected of a majority Islamic country, about 88% of the sample was Muslim.

In terms of missing observations, 26 people in 1993 were missing religion and 8 of them were missing education level. These were recoded to an 'other' category for regression analysis. Approximately 4% and 3% of rural women were breastfeeding at the time of the survey in 2000 and 2007, respectively. Among urban women, 4% and 2% were breastfeeding in 2000 and 2007, respectively. Urban and rural women had similar parities in 2000 and 2007: 2.8 live

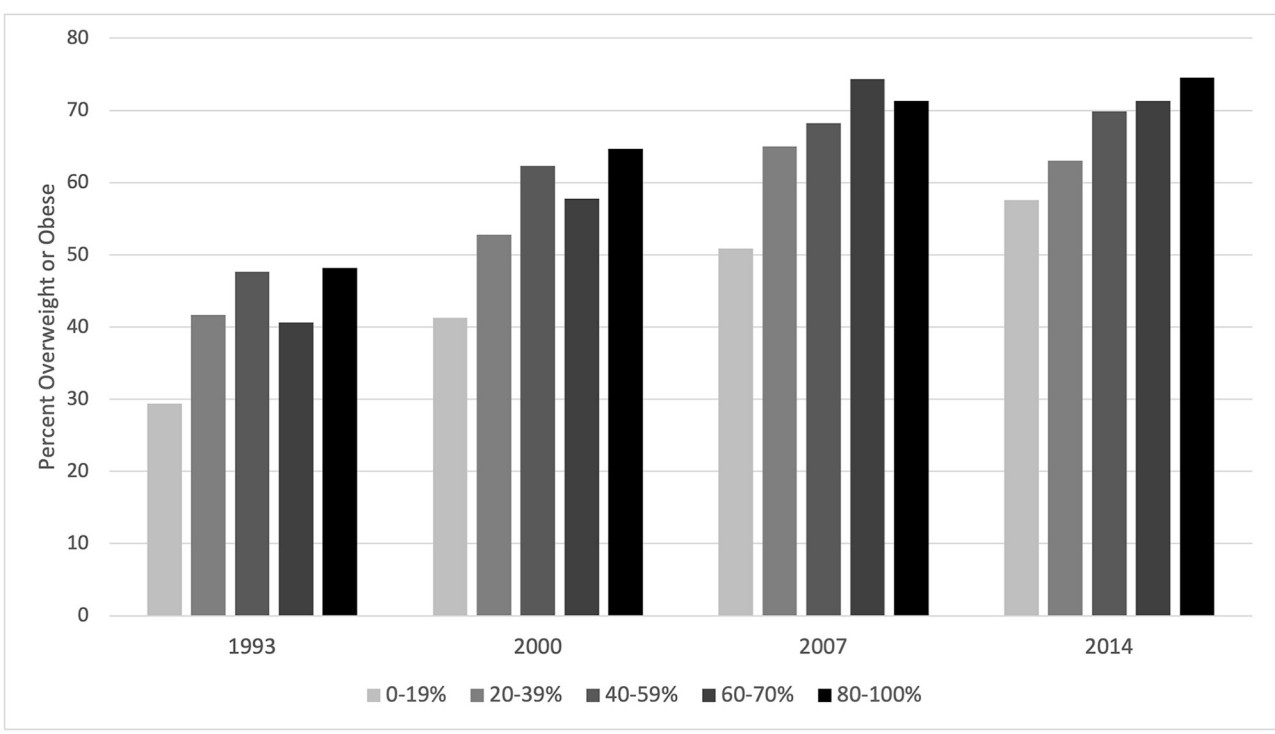

**Fig 5. Proportion overweight and obese by built-up categories, IFLS 1993–2014.**

births per woman in 2000 and 2.9 live births per woman in 2007. Only 3.8% of urban women but 56% of urban men smoked in 2007 and the proportions were higher for rural men and women. Most people did not move (>90% in all three periods). Among movers, urban-to-urban movers were the larger share. Since we only selected individuals who were interviewed in each panel, attrition bias was a concern. Fuwa found that attrition from a household panel had little quantitative importance for per capita household consumption in other similar surveys [30]. IFLS survey administrators suggest that reasons for individual movers are associated with several sociodemographic factors, some of which are not observed at baseline, and vary with the distance moved [31]. Therefore, we do not consider attrition (at household or individual levels) to be a significant factor in predicting BMI. Age, education, marital status, smoking, and religion were used as controls for regressions with 2000, 2007, and 2014 BMI outcomes.

Fig 5 shows the proportion of the population that is overweight or obese by quintiles of built-up area for each IFLS panel. The overwhelming trend is upward. Yet, there is a wide range in these proportions: in the most current wave, 58% of persons living in the least built-up area (less than 20% of the area is built up) are overweight or obese in comparison to 75% of those living in the most built-up area (more than 80% built up). For the other three panels, we observe similar differences by built-up area quintiles. Across survey waves, we see increasing fractions of overweight or obese, however, this may be due, in large part, to aging of individuals across the survey years, which is unaccounted here but adjusted for in the multivariate analysis below.

## Multivariate analysis

**Pooled sample.** We first discuss the results from pooled samples, combining the 1,464 men and 2,306 women respondents. Tables 3 and 4 show estimates of BMI regressions and Tables 5 and 6 show results for overweight/obese from linear probability regression models.

Table 3. Linear regression model predicting BMI, pooled sample.

| Variables | Model 1 (age +sex+ current urban/rural) | Model 2 (Model 1 + prior urban/rural) | Model 3 Model 2 + current built-up | Model 4 (Model 3 + prior built-up) | Model 5 (Model 3 + island) | Model 6 (Model 4 + island) | Model 7 (Model 5 + educ, married, smoker) | Model 8 (Model 6 + educ, married, smoker) |
|---|---|---|---|---|---|---|---|---|
| Percent built-up area of current Residence | | | 0.0222*** | | 0.0229*** | | 0.0184*** | |
| | | | (0.0018) | | (0.0019) | | (0.0019) | |
| Change in % built-up area since previous panel | | | | 0.0007 | | 0.0016 | | 0.0015 |
| | | | | (0.0014) | | (0.0015) | | (0.0014) |
| Percent built-up area of residence in previous panel | | | | 0.0229*** | | 0.0237*** | | 0.0192*** |
| | | | | (0.0019) | | (0.0020) | | (0.0020) |
| Current age | 0.1109*** | 0.1087*** | 0.1127*** | 0.1126*** | 0.1110*** | 0.1113*** | 0.0988*** | 0.0991*** |
| | (0.0263) | (0.0262) | (0.0264) | (0.0264) | (0.0263) | (0.0263) | (0.0261) | (0.0261) |
| Current age squared/100 | -0.1613*** | -0.1591*** | -0.1618*** | -0.1619*** | -0.1601*** | -0.1605*** | -0.1289*** | -0.1291*** |
| | (0.0235) | (0.0235) | (0.0236) | (0.0236) | (0.0236) | (0.0236) | (0.0234) | (0.0234) |
| Woman | 1.8726*** | 1.8670*** | 1.8791*** | 1.8791*** | 1.8735*** | 1.8740*** | 1.6702*** | 1.6704*** |
| | (0.1025) | (0.1024) | (0.1024) | (0.1024) | (0.1023) | (0.1023) | (0.1407) | (0.1407) |
| Island of residence (Reference = Java) | | | | | Ref | Ref | Ref | Ref |
| Sumatra | | | | | 0.2956** | 0.2941** | 0.1977 | 0.2009 |
| | | | | | (0.1448) | (0.1478) | (0.1458) | (0.1487) |
| All other islands | | | | | 0.0414 | 0.0489 | 0.0009 | 0.0109 |
| | | | | | (0.1335) | (0.1336) | (0.1462) | (0.1463) |
| Education (Reference = none) | | | | | | | Ref | Ref |
| Elementary | | | | | | | 0.9464*** | 0.9513*** |
| | | | | | | | (0.1393) | (0.1392) |
| Junior high | | | | | | | 1.2999*** | 1.3086*** |
| | | | | | | | (0.2009) | (0.2007) |
| Senior high | | | | | | | 1.959*** | 1.9690*** |
| | | | | | | | (0.2104) | (0.2101) |
| College or higher | | | | | | | 2.179*** | 2.1843*** |
| | | | | | | | (0.2713) | (0.2713) |
| Other | | | | | | | 1.2669*** | 1.278*** |
| | | | | | | | (0.2758) | (0.2755) |
| Marital status (Ref = Never married) | | | | | | | Ref | Ref |
| Married | | | | | | | 1.1363** | 1.1356** |
| | | | | | | | (0.5218) | (0.5246) |
| Widowed or other | | | | | | | 0.5706 | 0.5696 |
| | | | | | | | (0.5366) | (0.5394) |
| Religion (Reference = Islam) | | | | | | | Ref | Ref |
| Christianity | | | | | | | -0.1647 | -0.1656 |
| | | | | | | | (0.2226) | (0.2227) |
| Hindu, Buddhist, or other | | | | | | | 0.2307 | 0.2284 |
| | | | | | | | (0.2350) | (0.2350) |

(*Continued*)

**Table 3.** (Continued)

| Variables | Model 1 (age +sex+ current urban/rural) | Model 2 (Model 1 + prior urban/rural) | Model 3 Model 2 + current built-up | Model 4 (Model 3 + prior built-up) | Model 5 (Model 3 + island) | Model 6 (Model 4 + island) | Model 7 (Model 5 + educ, married, smoker) | Model 8 (Model 6 + educ, married, smoker) |
|---|---|---|---|---|---|---|---|---|
| Current smoker | | | | | | | -0.7919*** | -0.7935*** |
| | | | | | | | (0.1293) | (0.1294) |
| Period (Reference = 1993–2000) | Ref | Ref | Ref | Ref | Ref | Ref | Ref | Ref |
| 2000–2007 | 1.1779*** | 1.2162*** | 1.1703*** | 1.1823*** | 1.1688*** | 1.1702*** | 1.0628*** | 1.0610*** |
| | (0.0579) | (0.0575) | (0.0577) | (0.0595) | (0.0577) | (0.0593) | (0.0579) | (0.0595) |
| 2007–2014 | 2.0096*** | 2.1004*** | 2.0643*** | 2.0771*** | 2.0601*** | 2.0762*** | 1.8980*** | 1.9101*** |
| | (0.0914) | (0.0903) | (0.0906) | (0.0908) | (0.0905) | (0.0907) | (0.0922) | (0.0925) |
| Urban cluster (Reference = rural) | Ref | | | | | | | |
| Current urban strata | 1.2579*** | | | | | | | |
| | (0.1004) | | | | | | | |
| Previous wave urban strata | | 1.3463*** | | | | | | |
| | | (0.1041) | | | | | | |
| Observations (Persons) | 3,770 | 3,770 | 3,770 | 3,770 | 3,770 | 3,770 | 3,770 | 3,770 |
| $R^2$ | 0.149 | 0.153 | 0.153 | 0.153 | 0.153 | 0.153 | 0.182 | 0.182 |

(Robust standard errors in parentheses: *** $p<0.01$, ** $p<0.05$, * $p<0.1$).

The tables in the "b" series show the valued-added models discussed above as they add a person's lagged BMI to the specifications in Tables "a". Each table shows estimates from eight different regressions studying the relationship between physical built-up and weight, all adjusted for age, age squared, and period dummies (2007–2014, 2000–2007, and 1993–2000, which served as reference). Models 1 and 2 focus on the standard binary measure of urban vs. rural location provided in the IFLS, with Model 1 using current urban residence and Model 2 using lagged urban residence. Models 3 to 8 focus on our continuous measure of urbanicity, degree of built-up, using specifications with current built-up (Models 3, 5 and 7) or lagged built-up and period change in built-up (Models 3, 6 and 8), with different sets of controls.

Turning to the BMI regression results in Tables 3 and 4, we found that location of residence was an important predictor of BMI. Individuals in locations classified as urban had greater BMI values on average than rural residents. Based on Model 1 in Table 3, using current location data, the predicted urban-rural gap in mean BMI was 1.3 points. This is a significant difference both in statistical and practical terms. Using the lagged location (location 7 years prior) instead of the current one (contemporaneous to BMI measurement) returned a similar estimate of the urban-rural gap (see Model 2).

Looking at the estimates from specifications with our continuous measure of built-up percentage ("percent built-up area"), as shown in Models 3–8 in Table 3, we observed that the degree of built-up is strongly positively associated with BMI. For example, BMI values were about 0.22 BMI points higher on average in areas that were 10 percentage points more built up (based on Model 3).

The relationship between physical environment and BMI was robust to controls for island location (Sumatra or all others combined, versus Java as the reference), as shown in Model 5. However, it was reduced by 20% with the inclusion of socio-economic background variables

**Table 4. Value-added linear regression model predicting BMI, pooled sample.**

| Variables | Model 1 (age +sex+ current urban/rural) | Model 2 (Model 1 + prior urban/rural) | Model 3 Model 2 + current built-up | Model 4 (Model 3 + prior built-up) | Model 5 (Model 3 + island) | Model 6 (Model 4 + island) | Model 7 (Model 5 + educ, married, smoker) | Model 8 (Model 6 + educ, married, smoker) |
|---|---|---|---|---|---|---|---|---|
| Percent built-up area of current residence | | | 0.0043*** | | 0.0042*** | | 0.0031*** | |
| | | | (0.0007) | | (0.0007) | | (0.0007) | |
| Change in % built-up area since previous panel | | | | 0.0014** | | 0.0017** | | 0.0016** |
| | | | | (0.0007) | | (0.0007) | | (0.0007) |
| Percent built-up area of residence in previous panel | | | | 0.0045*** | | 0.0047*** | | 0.0036*** |
| | | | | (0.0007) | | (0.0008) | | (0.0008) |
| Current age | -0.0633*** | -0.0632*** | -0.0622*** | -0.0619*** | -0.0637*** | -0.0633*** | -0.0660*** | -0.0657*** |
| | (0.0155) | (0.0155) | (0.0155) | (0.0155) | (0.0156) | (0.0156) | (0.0156) | (0.0156) |
| Current age squared/100 | 0.0202 | 0.0201 | 0.0195 | 0.0192 | 0.0207 | 0.0204 | 0.0289** | 0.0286** |
| | (0.0145) | (0.0145) | (0.0145) | (0.0145) | (0.0145) | (0.0143) | (0.0145) | (0.0145) |
| Woman | 0.5201*** | 0.5206*** | 0.5229*** | 0.5231*** | 0.5195*** | 0.5196*** | 0.4622*** | 0.4625*** |
| | (0.0399) | (0.0399) | (0.0399) | (0.0399) | (0.0399) | (0.0399) | (0.0624) | (0.0624) |
| Island of residence (Ref = Java) | | | | | Ref | Ref | Ref | Ref |
| Sumatra | | | | | 0.0770 | 0.1039 | 0.0544 | 0.0818 |
| | | | | | (0.0548) | (0.0566) | (0.0562) | (0.0579) |
| All other islands | | | | | -0.0888* | -0.0767 | -0.0790 | -0.0683 |
| | | | | | (0.0490) | (0.0491) | (0.0547) | (0.0548) |
| Education (Ref = none) | | | | | | | Ref | Ref |
| Elementary | | | | | | | 0.2565*** | 0.2558*** |
| | | | | | | | (0.0619) | (0.0619) |
| Junior high | | | | | | | 0.3500*** | 0.3488*** |
| | | | | | | | (0.0853) | (0.0853) |
| Senior high | | | | | | | 0.6273*** | 0.6264*** |
| | | | | | | | (0.0888) | (0.0888) |
| College or higher | | | | | | | 0.6073*** | 0.6040*** |
| | | | | | | | (0.1016) | (0.1016) |
| Other | | | | | | | 0.3333** | 0.3276** |
| | | | | | | | (0.1488) | (0.1491) |
| Marital status (Ref = Never married) | | | | | | | Ref | Ref |
| Married | | | | | | | 0.5815** | 0.5817** |
| | | | | | | | (0.2476) | (0.2476) |
| Widowed or other | | | | | | | 0.3810 | 0.3782 |
| | | | | | | | (0.2544) | (0.2545) |
| Religion (Ref = Islam) | | | | | | | Ref | Ref |
| Christianity | | | | | | | -0.0647 | -0.0568 |
| | | | | | | | (0.0797) | (0.0799) |

*(Continued)*

**Table 4.** (Continued)

| Variables | Model 1 (age +sex+ current urban/rural) | Model 2 (Model 1 + prior urban/ rural) | Model 3 Model 2 + current built-up | Model 4 (Model 3 + prior built-up) | Model 5 (Model 3 + island) | Model 6 (Model 4 + island) | Model 7 (Model 5 + educ, married, smoker) | Model 8 (Model 6 + educ, married, smoker) |
|---|---|---|---|---|---|---|---|---|
| Hindu, Buddhist, or other | | | | | | | -0.0204 | -0.0146 |
| | | | | | | | (0.0888) | (0.0890) |
| Current smoker | | | | | | | -0.2603*** | -0.2607*** |
| | | | | | | | (0.0647) | (0.0648) |
| Period (Ref = 1993–2000) | Ref | Ref | Ref | Ref | Ref | Ref | Ref | Ref |
| 2000–2007 | 0.3758*** | 0.3843*** | 0.3751*** | 0.3620*** | 0.3776*** | 0.3609*** | 0.3551*** | 0.3380*** |
| | (0.0512) | (0.0512) | (0.0513) | (0.0516) | (0.0513) | (0.0516) | (0.0511) | (0.0514) |
| 2007–2014 | 0.3010*** | 0.3227*** | 0.3155*** | 0.3222*** | 0.3194*** | 0.3264*** | 0.2945*** | 0.3015*** |
| | (0.0521) | (0.0520) | (0.0521) | (0.0521) | (0.0522) | (0.0522) | (0.0532) | (0.0532) |
| Urban cluster (Ref = rural) | Ref | | | | | | | |
| Current urban strata | 0.2803*** | | | | | | | |
| | (0.0410) | | | | | | | |
| Previous wave urban strata | | 0.2716*** | | | | | | |
| | | (0.0416) | | | | | | |
| Lagged BMI | 0.8858*** | 0.8855*** | 0.8857*** | 0.8858*** | 0.8856*** | 0.8857*** | 0.8755*** | 0.8755*** |
| | (0.0104) | (0.0104) | (0.0104) | (0.0104) | (0.0104) | (0.0104) | (0.0106) | (0.0106) |
| Observations (Persons) | 3,770 | 3,770 | 3,770 | 3,770 | 3,770 | 3,770 | 3,770 | 3,770 |
| $R^2$ | 0.685 | 0.685 | 0.685 | 0.685 | 0.685 | 0.685 | 0.688 | 0.688 |

(Robust standard errors in parentheses: *** $p<0.01$, ** $p<0.05$, * $p<0.1$).

(educational attainment and marital status) and a dummy for being a smoker (Model 7). Adding these controls also improved the overall fit of the model as indicated by the higher $R^2$ (0.18 vs 0.15). The decline is consistent with the idea that socio-economic status and smoking are part of the mechanisms through which urbanization impacts weight.

Models 4, 6, and 8 in Table 3 report estimates for specifications using lagged built-up as a predictor along with the change in built-up during the preceding 7-year period. Lagged built-up percentage performed similarly to contemporaneous built-up in that it predicted BMI well and this association was robust to controls for island residence but diminished when adjusting for the potential confounders discussed above. The coefficient on change in built-up during a 7-year period between waves was estimated to be positive but not statistically significant. Conceptually, we would expect past built-up level (7-year lag) and subsequent change to both matter and, statistically, we found that they are in fact jointly significant. (As discussed above, this setup is mathematically equivalent to a specification with current and lagged built-up, which is a generalization of the model with only current built-up. The estimate on built-up *change* equals the coefficient on *current* built-up obtained from regressing BMI on current *and* lagged built-up.).

How does our built-up measure compare to the rural-urban dichotomy indicator? The results suggest that both measures perform similarly in terms of their contribution to explaining BMI: $R^2$ is 14.9% in the model with current urban-rural location dummy (Model 1) and 15.3% in the model with 7-year lagged urban dummy (Model 2) and in the models with

**Table 5. Linear regression model predicting overweight/obese, pooled sample.**

| Variables | Model 1 (age +sex+ current urban/rural) | Model 2 (Model 1 + prior urban/rural) | Model 3 Model 2 + current built-up | Model 4 (Model 3 + prior built-up) | Model 5 (Model 3 + island) | Model 6 (Model 4 + island) | Model 7 (Model 5 + educ, married, smoker) | Model 8 (Model 6 + educ, married, smoker) |
|---|---|---|---|---|---|---|---|---|
| Percent built-up area of current residence | | | 0.0025*** | | 0.0026*** | | 0.0020*** | |
| | | | (0.0002) | | (0.0002) | | (0.0002) | |
| Change in % built-up area since previous panel | | | | 0.0037* | | 0.0048** | | 0.0031 |
| | | | | (0.0020) | | (0.0020) | | (0.0020) |
| Percent built-up area of residence in previous panel | | | | 0.0025*** | | 0.0026*** | | 0.0020*** |
| | | | | (0.0002) | | (0.0002) | | (0.0002) |
| Current age | 0.0141*** | 0.0139*** | 0.0145*** | 0.0146*** | 0.0143*** | 0.0144*** | 0.0128*** | 0.0129*** |
| | (0.0033) | (0.0033) | (0.0034) | (0.0034) | (0.0034) | (0.0034) | (0.0033) | (0.0033) |
| Current age squared/100 | -0.0195*** | -0.0193*** | -0.0197*** | -0.0198*** | -0.0195*** | -0.0196*** | -0.0155*** | -0.0156*** |
| | (0.0031) | (0.0031) | (0.0031) | (0.0031) | (0.0031) | (0.0031) | (0.0031) | (0.0031) |
| Woman | 0.2196*** | 0.2190*** | 0.2207*** | 0.2208*** | 0.2200*** | 0.2201*** | 0.1709*** | 0.1710*** |
| | (0.0135) | (0.0135) | (0.0135) | (0.0135) | (0.0135) | (0.0136) | (0.0186) | (0.0186) |
| Island of residence (Ref = Java) | | | | | Ref | Ref | Ref | Ref |
| Sumatra | | | | | 0.0423** | 0.0434** | 0.0297* | 0.0316* |
| | | | | | (0.0177) | (0.0179) | (0.0177) | (0.0179) |
| All other islands | | | | | 0.0087 | 0.0097 | -0.0035 | -0.0022 |
| | | | | | (0.0168) | (0.0168) | (0.0182) | (0.0182) |
| Education (Ref = none) | | | | | | | Ref | Ref |
| Elementary | | | | | | | 0.1346*** | 0.1351*** |
| | | | | | | | (0.0186) | (0.0186) |
| Junior high | | | | | | | 0.1746*** | 0.1758*** |
| | | | | | | | (0.0245) | (0.0245) |
| Senior high | | | | | | | 0.2578*** | 0.2593*** |
| | | | | | | | (0.0260) | (0.0259) |
| College or higher | | | | | | | 0.3163*** | 0.3170*** |
| | | | | | | | (0.0334) | (0.0334) |
| Other | | | | | | | 0.1321*** | 0.1328*** |
| | | | | | | | (0.0340) | (0.0340) |
| Marital status (Ref = Nev. mar.) | | | | | | | Ref | Ref |
| Married | | | | | | | 0.0877 | 0.0875 |
| | | | | | | | (0.0816) | (0.0818) |
| Widowed or other | | | | | | | 0.0245 | 0.0242 |
| | | | | | | | (0.0833) | (0.0836) |
| Religion (Ref = Islam) | | | | | | | Ref | Ref |
| Christianity | | | | | | | -0.0167 | -0.0164 |
| | | | | | | | (0.0296) | (0.0296) |
| Hindu, Buddhist, or other | | | | | | | 0.0514* | 0.0517* |

*(Continued)*

**Table 5.** (Continued)

| Variables | Model 1 (age +sex+ current urban/rural) | Model 2 (Model 1 + prior urban/ rural) | Model 3 Model 2 + current built-up | Model 4 (Model 3 + prior built-up) | Model 5 (Model 3 + island) | Model 6 (Model 4 + island) | Model 7 (Model 5 + educ, married, smoker) | Model 8 (Model 6 + educ, married, smoker) |
|---|---|---|---|---|---|---|---|---|
| | | | | | | | (0.0302) | (0.0302) |
| Current smoker | | | | | | | -0.1367*** | -0.1370*** |
| | | | | | | | (0.0182) | (0.0182) |
| Period (Ref = 1993–2000) | Ref | Ref | Ref | Ref | Ref | Ref | Ref | Ref |
| 2000–2007 | **0.1291***** | **0.1337***** | **0.1285***** | **0.1293***** | **0.1282***** | **0.1274***** | **0.1131***** | **0.1118***** |
| | (0.0086) | (0.0086) | (0.0086) | (0.0089) | (0.0086) | (0.0089) | (0.0087) | (0.0089) |
| 2007–2014 | **0.2117***** | **0.2227***** | **0.2189***** | **0.2206***** | **0.2182***** | **0.2203***** | **0.1991***** | **0.2008***** |
| | (0.0119) | (0.0118) | (0.0118) | (0.0118) | (0.0119) | (0.0119) | (0.0121) | (0.0121) |
| Urban cluster (Ref = rural) | Ref | | | | | | | |
| Current urban strata | **0.1514***** | | | | | | | |
| | (0.0123) | | | | | | | |
| Prev. wave urban strata | | **0.1605***** | | | | | | |
| | | (0.0126) | | | | | | |
| Observations (Persons) | 3,770 | 3,770 | 3,770 | 3,770 | 3,770 | 3,770 | 3,770 | 3,770 |
| $R^2$ | 0.115 | 0.117 | 0.114 | 0.114 | 0.115 | 0.115 | 0.148 | 0.148 |

(Robust standard errors in parentheses: *** $p<0.01$, ** $p<0.05$, * $p<0.1$).

current (or lagged) built-up measures (Models 3 & 4). Evaluated at the average change in the proportion urban (+15.1 percentage points, based on Table 1) observed over the entire 21-year survey period, Model 1, suggests that the trend in urbanization as captured by the urban-rural dummy added 0.19 BMI points. Using our physical built-up measure, Model 3 predicts that the 6-percentage point average increase in built-up that occurred between 1993 and 2014 (see Table 1) translates into a BMI increase of 0.13 points (both calculations are based on models that adjust for age and sex).

A concern with the estimates in Table 3 is that they may over- or understate the contribution of period-specific determinants of weight such as built-up because of confounding with unmeasured past inputs and endowments in the weight determination process. The value-added regressions attempt to address this concern by conditioning on lagged BMI of the person, i.e., BMI in the previous panel. By doing this, we are focusing explicitly on the variation in individuals' weight during each 7-year period.

Table 4 shows the results of the value-added models that are based on the same eight specifications as in Table 3, but with lagged BMI added to each. As expected, these "value-added" specifications had a much better fit overall (higher $R^2$ values) as individuals' past BMI was highly predictive of their current BMI. The estimates associated with the built-up measures were smaller–and substantially so in some cases–than in Table 3. This pattern is consistent with the idea that the earlier estimates tended to overstate the built-up contribution to weight.

While the built environment was found to be a robust predictor of BMI across models, the implied contribution of built-up to overall, population-level BMI is rather small. For example, Model 5 in Table 4 suggests that a 10-percentage point increase in built-up between panel waves is associated with a 0.042 point rise in BMI. Since BMI values rose by about 0.8 points

**Table 6. Value-added linear regression model predicting overweight/obese, pooled sample.**

| Variables | Model 1 (age +sex+ current urban/rural) | Model 2 (Model 1 + prior urban/rural) | Model 3 Model 2 + current built-up | Model 4 (Model 3 + prior built-up) | Model 5 (Model 3 + island) | Model 6 (Model 4 + island) | Model 7 (Model 5 + educ, married, smoker) | Model 8 (Model 6 + educ, married, smoker) |
|---|---|---|---|---|---|---|---|---|
| Percent built-up area of current | | | **0.0008**\*\*\* | | **0.0008**\*\*\* | | **0.0005**\*\*\* | |
| Residence | | | (0.0001) | | (0.0002) | | (0.002) | |
| Change in % built-up area since previous panel | | | | 0.0002 | | **0.0003**\* | | **0.0003**\* |
| | | | | (0.0001) | | (0.0001) | | (0.0001) |
| Percent built-up area of residence in previous panel | | | | **0.0008**\*\*\* | | **0.0008**\*\*\* | | **0.0006**\*\*\* |
| | | | | (0.0002) | | (0.0002) | | (0.0002) |
| Current age | -0.0027 | -0.0027 | -0.0025 | -0.0024 | -0.0026 | -0.0026 | -0.0029 | -0.0028 |
| | (0.0025) | (0.0025) | (0.0025) | (0.0025) | (0.0025) | (0.0025) | (0.0025) | (0.0025) |
| Current age squared/ 100 | -0.0019 | -0.0019 | -0.0021 | -0.0022 | -0.0019 | -0.0020 | -0.0005 | -0.0005 |
| | (0.0023) | (0.0002) | (0.0023) | (0.0023) | (0.0023) | (0.0023) | (0.0023) | (0.0023) |
| Woman | **0.0885**\*\*\* | **0.0886**\*\*\* | **0.0890**\*\*\* | **0.0890**\*\*\* | **0.0885**\*\*\* | **0.0885**\*\*\* | **0.0556**\*\*\* | **0.0556**\*\*\* |
| | (0.0093) | (0.0093) | (0.0093) | (0.0093) | (0.0094) | (0.0094) | (0.0128) | (0.0128) |
| Island of residence (Ref = Java) | | | | | Ref | Ref | Ref | Ref |
| Sumatra | | | | | **0.0211**\* | **0.0249**\*\* | 0.0161 | **0.0202**\* |
| | | | | | (0.0118) | (0.0120) | (0.0119) | (0.0121) |
| All other islands | | | | | -0.0040 | -0.0025 | -0.0111 | -0.0098 |
| | | | | | (0.0109) | (0.0109) | (0.0120) | (0.0120) |
| Education (Ref = none) | | | | | | | Ref | Ref |
| Elementary | | | | | | | **0.0687**\*\*\* | **0.0687**\*\*\* |
| | | | | | | | (0.0129) | (0.0129) |
| Junior high | | | | | | | **0.0840**\*\*\* | **0.0842**\*\*\* |
| | | | | | | | (0.0170) | (0.0170) |
| Senior high | | | | | | | **0.1308**\*\*\* | **0.1311**\*\*\* |
| | | | | | | | (0.0184) | (0.0183) |
| College or higher | | | | | | | **0.1663**\*\*\* | **0.1662**\*\*\* |
| | | | | | | | (0.0236) | (0.0236) |
| Other | | | | | | | 0.0430 | 0.0421 |
| | | | | | | | (0.0281) | (0.0282) |
| Marital status (Ref = Never married) | | | | | | | Ref | Ref |
| Married | | | | | | | 0.0347 | 0.0347 |
| | | | | | | | (0.0606) | (0.0606) |
| Widowed or other | | | | | | | 0.0064 | 0.0059 |
| | | | | | | | (0.0679) | (0.0618) |
| Religion (Ref = Islam) | | | | | | | Ref | Ref |
| Christianity | | | | | | | -0.0071 | -0.0060 |
| | | | | | | | (0.0194) | (0.0194) |
| Hindu, Buddhist, or other | | | | | | | 0.0275 | 0.0285 |
| | | | | | | | (0.0203) | (0.0203) |

(*Continued*)

**Table 6.** (Continued)

| Variables | Model 1 (age +sex+ current urban/rural) | Model 2 (Model 1 + prior urban/rural) | Model 3 Model 2 + current built-up | Model 4 (Model 3 + prior built-up) | Model 5 (Model 3 + island) | Model 6 (Model 4 + island) | Model 7 (Model 5 + educ, married, smoker) | Model 8 (Model 6 + educ, married, smoker) |
|---|---|---|---|---|---|---|---|---|
| Current smoker | | | | | | | -0.0860*** | -0.0861*** |
| | | | | | | | (0.0126) | (0.0127) |
| Period (Ref = 1993–2000) | Ref | Ref | Ref | Ref | Ref | Ref | Ref | Ref |
| 2000–2007 | **0.0514***** | **0.0532***** | **0.0513***** | **0.0496***** | **0.0514***** | **0.0488***** | **0.0456***** | **0.0428***** |
| | (0.0086) | (0.0086) | (0.0086) | (0.0087) | (0.0086) | (0.0088) | (0.0086) | (0.0087) |
| 2007–2014 | **0.0461***** | **0.0506***** | **0.0491***** | **0.0500***** | **0.0492***** | **0.0504***** | **0.0461***** | **0.0472***** |
| | (0.0104) | (0.0104) | (0.0104) | (0.0104) | (0.0104) | (0.0104) | (0.0105) | (0.0105) |
| Urban cluster (Ref = rural) | Ref | | | | | | | |
| Current urban strata | **0.0567***** | | | | | | | |
| | (0.0086) | | | | | | | |
| Previous wave urban strata | | **0.0564***** | | | | | | |
| | | (0.0087) | | | | | | |
| Lagged BMI | **0.0858***** | **0.0857***** | **0.0860***** | **0.0861***** | **0.0860***** | **0.0860***** | **0.0835***** | **0.0836***** |
| | (0.0016) | (0.0016) | (0.0016) | (0.0016) | (0.0016) | (0.0016) | (0.0016) | (0.0016) |
| Observations (Persons = 3,770) | 3,770 | 3,770 | 3,770 | 3,770 | 3,770 | 3,770 | 3,770 | 3,770 |
| $R^2$ | 0.409 | 0.409 | 0.408 | 0.408 | 0.408 | 0.408 | 0.417 | 0.418 |

(Robust std. errors in parentheses: *** $p<0.01$, ** $p<0.05$, * $p<0.1$).

and built-up increased by about 2 percentage points on average in each 7-year period (see Table 1), this estimate suggests that a greater built environment contributed about 1.1% to the rise in mean BMI for the IFLS cohorts. (Based on Model 7 the contribution is 0.8%.) Given the structure of the value-added specification, this may represent a lower bound (i.e. conservative) estimate of the impact of an increase in built-up on weight.

The results from models of binary overweight/obese showed similar patterns as the analysis of mean BMI above. Looking at Tables 5 and 6, the standard measure of urban vs. rural location predicted the probability of being (Asian) overweight or obese well, with individuals in urban locations being more likely to be overweight or obese. Contemporaneous and lagged measures of the degree of built-up also predicted this risk well. Looking across Tables 5 and 6, change in built-up is statistically significant in Models 6 and 8 of the value-added specifications.

**Sex-stratified tables.** Looking at the stratified analysis by sex in S1–S8 Tables, we find that overall patterns are very similar among Indonesian men and women. Location and built-up land use tend to play similar roles for men and women for BMI and overweight/obesity. Generally, we found that the models were a better fit for men than women. Also, some statistically significant estimates in the pooled analysis were no longer significant in the stratified analysis, likely the result of the smaller sample sizes.

## Discussion

Indonesia has experienced rapid economic growth and urbanization in the past three decades. In this time, the prevalence of overweight and obesity has also doubled. We examined 21 years

of the IFLS panel data (1993–2014) to investigate the role of changing built environment on observed increases in mean BMI and proportion of individuals overweight and obese. We estimated longitudinal regression models using newly available matched geospatial measures of the percentage of land area that is built up. Three other studies have examined obesity trends in Indonesia using data from the same survey and found BMIs are rising among the entire population, particularly among urban women [6, 32, 33]. Our study built on those findings by expanding the focus explicitly to the role of urbanization using satellite data to classify built-up, and analyzing change over time for individual BMI observations.

We found that living in more built-up areas was associated with greater BMI and risk of becoming overweight or obese. The effect sizes associated with the built environment were estimated to be small but statistically significant even in our most conservative models that accounted for individuals' initial BMI. The results suggest that urbanization, as captured by a 6%-point average increase in built-up land area, accounted for a relatively small portion (around 1%) of the rise in overweight and obesity in Indonesia. The contribution is similar to using the urban-rural dichotomous measure available in the IFLS survey.

To put these changes in context, 50% built-up thresholds are used in global work as an indicator of urbanization, but suburban locations are characterized by much lower levels of built-up area (for example, as low as 15% in the US) while city centers of large urban areas are much higher (typically close to 80% or more on average in the United States) [15, 34]. A recent study of Indonesia's Sulawesi Province using built-up data from GHSL finds urbanization unfolding outside areas officially designated with implications not only for the provision of municipal services but also the potential to misclassify relevant aspects of the health and well-being [35]. A recent study in China that evaluated urbanization trajectories found a higher risk of being overweight and obese among men in areas with more urbanized features with no differences observed among women by urbanization trajectories [36]. As urbanization proceeds in Indonesia, it is expected that built-up percentages will increase much more than the 6%-point change we observed in the past which poses an increased risk of overweight and obesity over time. China and India provide different models of potential built-up related BMI outcomes in Indonesia. As we noted in India, city size also matters, and small towns or suburbs may eventually see additional risk as they undergo urban and nutrition transitions related to their proximity to larger urban areas [9, 37]. Our understanding of the drivers of overweight and obesity over time in the context of unplanned and rapid urbanization in low and middle-income countries is still evolving [38]. While neighborhood walkability, crime, and access to green spaces and healthy foods are known correlates of obesity among adults and children in western nations [39, 40], country-appropriate measures of similar determinants of health in poorer nations are still lacking. In a global study of urban walkability, Indonesia's suburbs, like many in South and Southeast Asia, are noted for their sprawling expanses, lack of sidewalks, and development of gate communities associated with an increasing middle-class, factors all of which contribute to a lack of walkability [41]. Further, a recent analysis using GHSL built-up finds that increases in density lead to lower levels of social capital, in particular trust in one's neighbors and participation in the community [42]. Future work on health outcomes should formally examine the role of these factors for Indonesia, including the use of fine-resolution census and other spatial data that were not available for the current study.

Finally, in Indonesia, the fast pace of urbanization has not been matched with sufficient infrastructure and services [43]. According to the World Bank, although the country's economy grew by an average of 5.8% in the mid to late 2000s, in the wake of the Asian financial crisis only 3% of GDP was invested in infrastructure per year on average, compared to 10% in China [44]. For example, due to inadequate transportation, commutes in Jakarta can average up to 2.5 hours regardless of public or private transport [45]. Traditionally the country has

been divided into *kota* (municipalities) and *kabupaten* (non-urban districts). However, since 1990, urban populations have grown faster in *kabupaten* than in *kota*, and within the *kabupaten* non-statutory towns are common. This is a challenge as the governance structure in a *kabupaten* is often not equipped to manage urban development. Insufficient infrastructure can have immediate impacts on health through lack of access to safe drinking water and sewage removal, but also chronic diseases and obesity through mechanisms such as neighborhoods with poor walkability and inadequate green spaces [46]. Rapidly developing countries may miss out on opportunities for healthy urban planning if urbanization outpaces infrastructure spending.

## Strengths and limitations

The present study had several strengths worth noting. First, we employed a novel, direct measure of built-up area, which was able to distinguish urban residential areas with more granularity than administrative or other definition-based boundaries, and which could capture aspects of the built environment that affect BMI-related behaviors such as walking. These alternative measures build on a more traditional urban-rural comparison. Second, we used individual-level panel data, which allowed us to examine weight trajectories over long periods and to control for individuals' initial (lagged) BMIs. The results confirmed the importance of individual heterogeneity in the weight determination process and suggest that studies that fail to account for this may overestimate the contribution of background factors such as urbanization. Finally, BMI mismeasurement was not a concern here since weight and height were measured by health professionals rather than self-reported [47–49].

Given the limited set of consistently measured variables available in the IFLS, we could not carefully examine specific causal pathways between urbanization and obesity. Several plausible explanations have been put forth. Economic development and technological change can result in weight gain by lowering calorie prices and rising incomes, making it affordable to increase calorie consumption, and by increasing the opportunity cost of meal preparation at home–resulting in the consumption of calorically denser food [50]. Some argue that a nutrition transition is well underway in developing countries such that processed foods rich in calories from fat and sugar have become more widely available, particularly in urban areas [7]. By 2030, the combined effect of wider availability of fast-food products, higher caloric intake from refined and processed foods, and sedentary work and life conditions associated with urban living, could contribute to a 75% increase in the prevalence of overweight and obesity among adults ages 20 years and older worldwide [51].

Food and built environments in cities elsewhere have been shown to interact to produce 'obesogenic' environments that can spatially pattern higher prevalence of overweight and obesity [52]. Greater access to local supermarkets versus neighborhood convenience stores was associated with lower BMI in American urban adolescents, especially among those with higher SES [53]. Another possibly important mechanism is decreasing calorie expenditure through less physical activity. In the developed world, urban sprawl and suburbanization after World War II led to increased reliance on vehicular transport instead of walking or biking and limited physical activity at work and home, with corresponding increases in BMI [54, 55].

Less is known about the ways the built environment affects physical activity in the developing world. Given that our measures of urbanization remained important predictors even after accounting for socio-economic factors, the results are consistent with the presence (and potential longer-term impact) of urban-rural differences in physical activity. This interpretation is in line with an overview of DHS surveys that found a link between urbanization and more sedentary lifestyles in developing countries [56]. In major cities across Ghana, Zimbabwe,

Bulgaria, and Nigeria, urban residents have become more vulnerable to unhealthy weight gain, in part due to macro-level economic trends that promote the consumption of energy-dense processed foods and sedentary working conditions [57–59].

## Conclusions

Simple urban-rural dichotomies that are generated from sampling frames tell us little about why some health outcomes are worse in urban settings. Satellite-derived spatially-oriented proxy measures should be able to tell us more about the character, size, and density of those urban settings, as well as potential commuting needs within those settings (for example living in core vs. peri-urban areas). However, as we found, satellite measures of built-up do not completely capture the nature of human interaction with their built environment. Additional research is needed to determine the optimal way to use GHS or similar spatial data sets that characterize the built environment and measure how people navigate it (such as the use of walking and common transportation routes) to more deeply examine the role of urbanization on BMI and other health effects. Additional measures of urbanization that can be generated through satellite measures and importantly, be linked to economic development (such as by use of night-time lights or road data, or data estimating the vertical dimensions of urbanization) would be important to understand changes in population health in the developing world. Furthermore, an exploration of access to safe public spaces for physical activity given cultural restrictions based on sex, ethnicity, or religious identity is warranted in countries such as Indonesia where public and private spaces may be inequitably accessible. Obesity is recognized as a major public health problem in wealthier countries. This study has unambiguously shown that rising BMI values and rising proportions of overweight or obese are major concerns in poorer countries like Indonesia, becoming only more important as urbanization unfolds. This will ultimately result in higher rates of chronic disease.

## Supporting information

**S1 Table. Linear regression model predicting BMI, women's sample.**
(DOCX)

**S2 Table. Value-added linear regression model predicting BMI, women's sample.**
(DOCX)

**S3 Table. Linear regression model predicting overweight/obese, women's sample.**
(DOCX)

**S4 Table. Value-added linear regression model predicting overweight/obese, women's sample.**
(DOCX)

**S5 Table. Linear regression model predicting BMI, men's sample.**
(DOCX)

**S6 Table. Value-added linear regression model predicting BMI, men's sample.**
(DOCX)

**S7 Table. Linear regression model predicting overweight/obese, men's sample.**
(DOCX)

**S8 Table. Value-added linear regression model predicting overweight/obese, men's sample.**
(DOCX)

## Acknowledgments

We wish to acknowledge the CUNY Institute for Demographic Research for supporting AD and JB as Demography Fellows during the process of manuscript development.

## Author Contributions

**Conceptualization:** Alka Dev, Jennifer Brite, Frank W. Heiland, Deborah Balk.

**Data curation:** Alka Dev, Jennifer Brite.

**Formal analysis:** Alka Dev, Jennifer Brite, Frank W. Heiland, Deborah Balk.

**Methodology:** Alka Dev, Jennifer Brite, Frank W. Heiland, Deborah Balk.

**Supervision:** Alka Dev, Frank W. Heiland, Deborah Balk.

**Visualization:** Deborah Balk.

**Writing – original draft:** Alka Dev, Jennifer Brite, Frank W. Heiland, Deborah Balk.

**Writing – review & editing:** Alka Dev, Jennifer Brite, Frank W. Heiland, Deborah Balk.

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
