## [Decision Letter · Decision Letter 0]

31 Jan 2022

PGPH-D-21-01064

Built environment as a risk factor for adult overweight and obesity: evidence from a longitudinal geospatial analysis in Indonesia

Dear Dr. Dev,

Thank you for submitting your manuscript to PLOS Global Public Health. After careful consideration, we feel that it has merit but does not fully meet PLOS Global Public Health’s publication criteria as it currently stands. Therefore, we invite you to submit a revised version of the manuscript that addresses the points raised during the review process.

We look forward to receiving your revised manuscript.

Kind regards,

Giridhara Rathnaiah Babu, MBBS, MPH, PhD

Academic Editor

Journal Requirements:

1. Please provide separate figure files in .tif or .eps format only, and remove any figures embedded in your manuscript file.

2. We have noticed that you have uploaded supporting information but you have not included a list of legends.  Please add a full list of legends for all supporting information files (including figures, table and data files) after the references list. 

3. In the online submission form, you indicated that "The data that support the findings of this study are available from RAND Corp. at [https://www.rand.org/well-being/social-and-behavioral-policy/data/FLS/IFLS.html]. GPS locations for IFLS clusters are IRB-protected and not publicly available due to information that could compromise the privacy of communities still participating in the survey. They could be requested from RAND Corp. upon institutional IRB approval.". All PLOS journals now require all data underlying the findings described in their manuscript to be freely available to other researchers, either 1. In a public repository, 2. Within the manuscript itself, or 3. Uploaded as supplementary information.

4. Please provide us with a direct link to the base layer of the map used in Fig 2 and ensure this location is also included in the figure legend. 

Please note that, because all PLOS articles are published under a CC BY license (creativecommons.org/licenses/by/4.0/), we cannot publish proprietary maps such as Google Maps, Mapquest or other copyrighted maps. If your map was obtained from a copyrighted source please amend the figure so that the base map used is from an openly available source.

Please note that only the following CC BY licences are compatible with PLOS licence: CC BY 4.0, CC BY 2.0  and CC BY 3.0, meanwhile such licences as CC BY-ND 3.0 and others are not compatible due to additional restrictions. If you are unsure whether you can use a map or not, please do reach out and we will be able to help you. 

The following websites are good examples of where you can source open access or public domain maps:

5. Please amend your detailed Financial Disclosure statement. This is published with the article, therefore should be completed in full sentences and contain the exact wording you wish to be published.

ii). State the initials, alongside each funding source, of each author to receive each grant.

iii). State what role the funders took in the study. If the funders had no role in your study, please state: “The funders had no role in study design, data collection and analysis, decision to publish, or preparation of the manuscript.”

iv). If any authors received a salary from any of your funders, please state which authors and which funders.

Additional Editor Comments (if provided):

Reviewers' comments:

Reviewer's Responses to Questions

**Comments to the Author**

1. Does this manuscript meet PLOS Global Public Health’s publication criteria? Is the manuscript technically sound, and do the data support the conclusions? The manuscript must describe methodologically and ethically rigorous research with conclusions that are appropriately drawn based on the data presented.

Reviewer #1: No

Reviewer #2: Yes

2. Has the statistical analysis been performed appropriately and rigorously?

Reviewer #1: No

Reviewer #2: No

3. Have the authors made all data underlying the findings in their manuscript fully available (please refer to the Data Availability Statement at the start of the manuscript PDF file)?

Reviewer #1: Yes

Reviewer #2: Yes

4. Is the manuscript presented in an intelligible fashion and written in standard English?

Reviewer #1: No

Reviewer #2: Yes

5. Review Comments to the Author

Reviewer #1: This paper explores the association of built environment with adults’ overweight and obesity in Indonesia. There are several problems in the analytical method, variable selection, and the contribution of the paper is unclear, which need to be further modified.

1. The contribution of this paper is unclear. Previous studies have widely explored the relationship between built environment and obesity, and have found that the built environment contributes to obesity. I do not see any new finding from this study. How does this paper contribute to the existing literature? Please clarify.

2. This paper was conducted in Indonesia, which is a country rarely concerned by previous studies. Therefore, I wonder if Indonesia has distinct characteristics which may influence the relationship between built environment and obesity. As a developing country, does the built environment and the obesity prevalence in Indonesia differ from those in developed countries (e.g., the U.S.)? Moreover, as an island country, does its built environment differ from that in continental states (e.g., China, India)? Please pay more attention to the local context of Indonesia. I would like to see more interesting results.

3. Considering that the longitudinal data is collected, why not apply cross-lagged panel models with fixed effects? I suggest the authors can apply the fixed effects models or cross-lagged panel models with fixed effects.

4. In the paper, the authors applied the dichotomous variable to distinguish between urban and rural areas. I wonder what are the differences between urban and rural built environment in details? Are there any different associations of the built environment with obesity found between urban and rural areas? Please describe the specific criteria for the division of urban and rural areas and explain the reasons for selecting the criteria.

5. Spatial analysis is based on a 2-km circular buffer around each cluster’ centroid. First, I wonder what is the cluster? Is it a neighborhood or a respondent’s house? Please clarify. Second, why do you only select 2 kilometers as the buffer radius? This choice should be underpinned. However, I suggest you can try more buffers with different radii as the unit of spatial analysis. Third, why do you use the circular buffer instead of the network buffer? As far as I know, a network buffer can better reflect the built environment exposed by individuals.

6. The logic of the paper is not flow. Many typos. The paper needs to be proofread.

Reviewer #2: This study used panel data and a 2km radius of built-up information of respondents' homes between 1993 and 2014 to investigate the impact of a built environmental attribute on the overweight and obesity of Indonesian adults. The study is relevant to the high prevalence of overweight and obesity worldwide. In particular, studies from developing countries are limited. The following lists a few issues that should be addressed before this paper is accepted for publication.

1. It is unclear why a 2 km radius of the respondent's home was used to measure the built-up area.

2. More importantly, the authors only used fixed-effect models. Given the nature of panel data, mixed models could be applied here. However, it is unclear why the authors did not use them.

6. PLOS authors have the option to publish the peer review history of their article (what does this mean?). If published, this will include your full peer review and any attached files.

**Do you want your identity to be public for this peer review?** For information about this choice, including consent withdrawal, please see our Privacy Policy.

Reviewer #1: No

Reviewer #2: No

---

## [Decision Letter · Decision Letter 1]

27 May 2022

PGPH-D-21-01064R1

Built environment as a risk factor for adult overweight and obesity: evidence from a longitudinal geospatial analysis in Indonesia

Dear Dr. Dev,

Thank you for submitting your manuscript to PLOS Global Public Health. After careful consideration, we feel that it has merit but does not fully meet PLOS Global Public Health’s publication criteria as it currently stands. Therefore, we invite you to submit a revised version of the manuscript that addresses the points raised during the review process.

We look forward to receiving your revised manuscript.

Kind regards,

Giridhara Rathnaiah Babu, MBBS, MPH, PhD

Academic Editor

Journal Requirements:

1. Please amend your Financial Disclosure statement. If you did not receive any funding for this study, please simply state: “The authors received no specific funding for this work.”

2. Please update your Competing Interests statement. If you have no competing interests to declare, please state: “The authors have declared that no competing interests exist.”

3. We noticed that you used “data not shown”/“not shown” in the manuscript. We do not allow these references, as the PLOS data access policy requires that all data be either published with the manuscript or made available in a publicly accessible database. Please amend the supplementary material to include the referenced data or remove the references.

4. Please ensure that all Figure files have corresponding citations and legends within the manuscript. Currently, Figure 1 in your submission file inventory does not have an in-text citation. If the figure is no longer to be included as part of the submission, please remove it from the file inventory.

Additional Editor Comments (if provided):

Reviewers' comments:

Reviewer's Responses to Questions

**Comments to the Author**

1. If the authors have adequately addressed your comments raised in a previous round of review and you feel that this manuscript is now acceptable for publication, you may indicate that here to bypass the “Comments to the Author” section, enter your conflict of interest statement in the “Confidential to Editor” section, and submit your "Accept" recommendation.

Reviewer #1: (No Response)

Reviewer #3: All comments have been addressed

Reviewer #4: All comments have been addressed

Reviewer #5: All comments have been addressed

2. Does this manuscript meet PLOS Global Public Health’s publication criteria? Is the manuscript technically sound, and do the data support the conclusions? The manuscript must describe methodologically and ethically rigorous research with conclusions that are appropriately drawn based on the data presented.

Reviewer #1: Partly

Reviewer #3: Yes

Reviewer #4: Yes

Reviewer #5: Yes

3. Has the statistical analysis been performed appropriately and rigorously?

Reviewer #1: No

Reviewer #3: I don't know

Reviewer #4: Yes

Reviewer #5: Yes

4. Have the authors made all data underlying the findings in their manuscript fully available (please refer to the Data Availability Statement at the start of the manuscript PDF file)?

Reviewer #1: No

Reviewer #3: Yes

Reviewer #4: Yes

Reviewer #5: Yes

5. Is the manuscript presented in an intelligible fashion and written in standard English?

Reviewer #1: Yes

Reviewer #3: Yes

Reviewer #4: Yes

Reviewer #5: Yes

6. Review Comments to the Author

Reviewer #1: I appreciate the author’s efforts to improve this paper. However, they did not address most of my concerns. The paper needs to be revised substantially and I do not satisfy with the current version.

1.The authors argued that the longitudinal data are a major contribution of this study. However, they do not consider fixed effects model or other models for panel data. Thus, their results do not have new contributions compared to the cross-sectional design. They argued that fixed effects models are very inefficient. However, the fixed effect models can remove the effects of time-invariant variables, providing more accurate estimations.

2. I believe developing countries (e.g. Indonesia) are important. In particular, the authors argued that “part of the contribution of this paper is to compare how commonly used indicators of urbanization (based on simple urban-rural dichotomies) perform in comparison to more sophisticated measures of urbanization (based on built-up).” However, the authors need to highlight the NEW findings from Indonesia and explain why these findings are different from other contexts. At least, I do not see any NEW finding in this version.

3. The 2km (or 5km, 10km) buffers are too large because people’s walking distances are about 1km. Most previous studies use 1/4 or 1/2 miles buffers.

Reviewer #3: This paper addresses an important topic using longitudinal data.

Page 2, line 9: "gross domestic product (GDP), has doubled from $4.8K to $11.1K". Please specify the currency and what "K" stands for.

Page 6, Line 123: "Two trained nurses assessed all individuals for health measurements during the survey, unless participants were too ill or pregnant". What classify as "too ill"? Are pregnant women excluded from analysis? How is pregnancy-related (including post pregnancy) overweight and obesity controlled in this study?

Inconsistent terminology. Throughout the document the terms gender/sex, men/male and women/female are used interchangeably (such as Page 43 line 458 "Gender-stratified tables"). However, I believe the data set uses biological sex, not gender.

Table 3a - 4b are confusing. instead of Models 1 - 8, are you able to change the headers that reflect the actual models?

The many tables on sociodemographic factors are fascinating not that necessary. Since this paper focuses on built environment, can you omit some of the information that's not too relevant?

Reviewer #4: This is a well structured and written paper, on a topic that is important to today’s pressing public health challenges. I thought that the background section provided helpful insight into the topic and some limitations that some methods for assessing the BMI-obesity-built environment relationship present.

My major comment is on a few variables that are considered important for this relationship but were missing in your discussion. In the discussion section on line 497, you mention that your understanding of the drivers of overweight and obesity over time in the context of unplanned and rapid urbanization is still evolving, however, I urge you to review the literature on how neighborhood crime, walkability scores, bike-ability, perceptions of neighborhood safety, access to public transportation, density of amenities and street lighting, and how these potentially affect physical activity.

I am also curious on how your analysis considered participant migration in and out of their primary areas of residence. Is it possible that for some participants, they did not live fully in their primary locations as reported by the data? I may have missed this in your discussion of the data, but if otherwise, then this is a factor that could greatly skew the results.

Reviewer #5: The background Objective, methods and findings were adequate. However, there is need to recast the last paragraph and include what the result/what was found mean. What are the authors implying with what has been seen in the study? What is the implication on the health of the urban population?

7. PLOS authors have the option to publish the peer review history of their article (what does this mean?). If published, this will include your full peer review and any attached files.

**Do you want your identity to be public for this peer review?** For information about this choice, including consent withdrawal, please see our Privacy Policy.

Reviewer #1: No

Reviewer #3: No

Reviewer #4: No

Reviewer #5: No

---

## [Decision Letter · Decision Letter 2]

7 Sep 2022

Built environment as a risk factor for adult overweight and obesity: evidence from a longitudinal geospatial analysis in Indonesia

PGPH-D-21-01064R2

Dear Dr Dev,

We are pleased to inform you that your manuscript 'Built environment as a risk factor for adult overweight and obesity: evidence from a longitudinal geospatial analysis in Indonesia' has been provisionally accepted for publication in PLOS Global Public Health.

Best regards,

Giridhara R Babu, MBBS, MPH, PhD

Academic Editor

Reviewer Comments (if any, and for reference):

Reviewer's Responses to Questions

**Comments to the Author**

1. If the authors have adequately addressed your comments raised in a previous round of review and you feel that this manuscript is now acceptable for publication, you may indicate that here to bypass the “Comments to the Author” section, enter your conflict of interest statement in the “Confidential to Editor” section, and submit your "Accept" recommendation.

Reviewer #1: All comments have been addressed

Reviewer #4: All comments have been addressed

2. Does this manuscript meet PLOS Global Public Health’s publication criteria? Is the manuscript technically sound, and do the data support the conclusions? The manuscript must describe methodologically and ethically rigorous research with conclusions that are appropriately drawn based on the data presented.

Reviewer #1: Yes

Reviewer #4: Yes

3. Has the statistical analysis been performed appropriately and rigorously?

Reviewer #1: Yes

Reviewer #4: Yes

4. Have the authors made all data underlying the findings in their manuscript fully available (please refer to the Data Availability Statement at the start of the manuscript PDF file)?

Reviewer #1: No

Reviewer #4: Yes

5. Is the manuscript presented in an intelligible fashion and written in standard English?

Reviewer #1: Yes

Reviewer #4: Yes

6. Review Comments to the Author

Reviewer #1: I do not have additiaonl comments. The journal requires that "in addition to summary statistics, the data points behind means, medians and variance measures should be available."

Reviewer #4: Comments provided in attachment

7. PLOS authors have the option to publish the peer review history of their article (what does this mean?). If published, this will include your full peer review and any attached files.

**Do you want your identity to be public for this peer review?** For information about this choice, including consent withdrawal, please see our Privacy Policy.

Reviewer #1: No

Reviewer #4: **Yes: **Sarah Lebu
